# The Importance of Implementing Cyber Physical Systems to Acquire Real-Time Data and Indicators

**Paula Morella** [1,*], **María Pilar Lambán** [1], **Jesús Antonio Royo** [1] **and Juan Carlos Sánchez** [2]

1 Design and Manufacturing Engineering Department, Universidad de Zaragoza, 50018 Zaragoza, Spain; plamban@unizar.es (M.P.L.); jaroyo@unizar.es (J.A.R.)
2 Smart Systems, Tecnalia, 20009 Donostia-San Sebastian, Spain; jcarlos.sanchez@tecnalia.com
* Correspondence: 620453@unizar.es

**Abstract:** Among the new trends in technology that have emerged through the Industry 4.0, Cyber Physical Systems (CPS) and Internet of Things (IoT) are crucial for the real-time data acquisition. This data acquisition, together with its transformation in valuable information, are indispensable for the development of real-time indicators. Moreover, real-time indicators provide companies with a competitive advantage over the competition since they enhance the calculus and speed up the decision-making and failure detection. Our research highlights the advantages of real-time data acquisition for supply chains, developing indicators that would be impossible to achieve with traditional systems, improving the accuracy of the existing ones and enhancing the real-time decision-making. Moreover, it brings out the importance of integrating technologies 4.0 in industry, in this case, CPS and IoT, and establishes the main points for a future research agenda of this topic.

**Keywords:** real time; data acquisition; Cyber Physical Systems; Internet of Things; Industry 4.0

## 1. Introduction

Industry 4.0 is the recent revolution of the industrial environment. This revolution merges the Internet, and information and communication technologies (ICT) with traditional manufacturing processes [1]. The aim of Industry 4.0 is the development of smart factories through the digitalization of manufacturing and assembly process [2]. Cyber Physical Systems (CPS) stand out from the other technologies of Industry 4.0 due to their high levels of control, surveillance, transparency, and efficiency in production process [3]. CPS can be defined as the integration between computation and physical processes [4]. This integration allows the real-time data acquisition. This acquisition is stored and transmitted to other devices thanks to the Internet of Things (IoT). Finally, these data can be processed to obtain valuable information and Key Performance Indicators (KPIs), enhancing the real-time decision-making in Supply Chains (SC) [5] (see Figure 1).

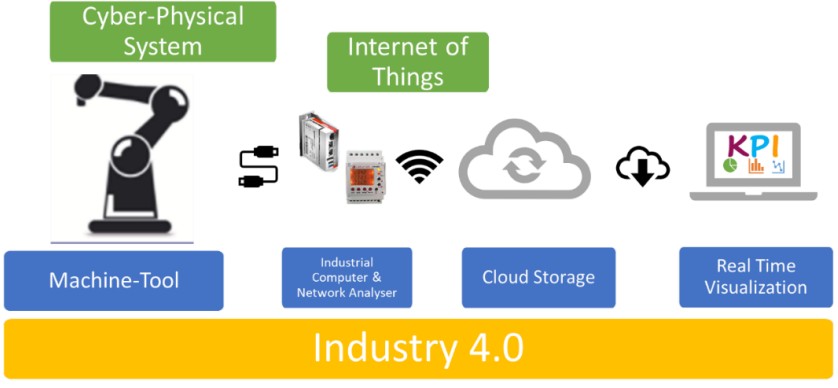

**Figure 1.** Integration for real-time acquisition.

Industry 4.0 enhances the quick and flexible reaction of companies to market changes. Particularly, the use of CPS end with the rigid structures within supply chains [6], enabling the monitorization and acquisition of real-time information for improving production scheduling, integrating supply chain partners in real time, reducing environmental uncertainty, and improving SC efficiency [7]. Moreover, changes can be detected earlier by observing the profitability development over time [5].

A real-time acquisition means an accurate data acquisition and therefore, a more accurate calculation of KPIs. For example, CPS enhances the acquisition and use of time variables to obtain more accurate KPIs than before, given that the use of average times and ratios are not necessary. This accuracy is interesting for cost indicators, which are usually calculated from historical data and cannot represent current reality. In the same way, CPS enables the monitorization of the energy consumption [8,9]. This monitorization can be used to develop KPIs based on energy efficiency, which are increasing their importance, since, the 70% of energy consumption in industries belongs to the machines [10]. It can be also considered as a predictive maintenance indicator; if there is a great increase in the energy consumption value, it can be alerted in real time and the machine should be reviewed to check if something is going wrong.

Implementing the real-time KPIs' calculation in CPS can reduce, or even eliminate, complex calculus and enable real-time responses to unforeseen failures. Once the KPIs equations are developed and implemented, thanks to the digitalization, the calculation in real time is immediate. Furthermore, the systems can be programed to display the information as visually as possible, informing us at a glance about the current situation of our machines [11]. This real-time information allows us to generate improvement actions and evaluate the changes in our supply chain and decision-making in real time.

To highlight the importance of this real-time information, this paper shows a case study in a Cyber Physical System, which is able to acquire the diary energy consumption per tool and to show it to the users. It allows the users the analysis of the machine-tools situation and the detection of energy improvements in regard to machine-tools. Furthermore, it can be seen what percentage of total energy consumption is associated with machine tools.

This manuscript is organized as follows: Section 2 describes some related works on this topic; Section 3 presents the case study and how the calculus and KPIs are developed. Section 4 shows the results of this implementation, which are discussed in Section 5. The paper ends with conclusions and proposals for future studies.

## 2. Related Works

Due to the advantages which can be achieved using real-time data, supply chains and industries are trying to take advantage of the sensors and actuators. Nowadays, these devices are crucial in supply chains and processes, which are impossible to monitor or control by humans because of their complexity [12]. The combination of the IoT and CPS is a key feature of Industry 4.0 [13]. Thanks to these technologies, the systems can accomplish their tasks based on information coming from the physical and virtual world [14]. The number of interconnected devices is increasing sharply given the potential application of these technologies in different industries, i.e., the supply chain of the manufacturing sector, engineering, finance, health sector, etc. [15]. Some of the applications of real- time data acquisition in different industries are shown in this section.

Electric Vehicles (EV) can use online environmental data to estimate the real-time state of charge and the remaining range for the EV while on the road [16]. Another use related to energy efficiency can be seen in this research, in which a prototype uses real-time indoor thermal information and real-time weather information together with user's body temperature to enhance the energy efficiency in buildings [17]. In terms of transport, real-time data can be used to correct the estimation made in origin-destination travel matrices for public transport using real-time and historical data [18]. These technologies enhance the implementation of advanced control strategies, e.g., it is used in chemical supply chains to challenge the fermentation processes which are complex and variable [19].

As can be seen, one of the most used cases is related to the environment and sustainability in different sectors. However, the development of energy efficiency in industry still has a generalized lack of awareness, knowledge, and experience on how to implement these concepts [20]. The energy costs and the huge amount of environmental legislatives and social requirements highlights that more efficient machine tools are required [21], particularly, the spindle use must be attended because it has a dominant influence on the energy demand [22]. For that reason, this research is focused on the energy consumption per tool, which is allocated in the spindle.

## 3. Materials and Methods

Our case study has been implemented in a CPS, whose physical system consists of a five-axis vertical milling machine (HAAS VF-3); two of these 5 axes were added by the incorporation of a Trunnion 160 double-cradle-table. For the cybernetic system, an industrial computer, which acquires the system variables, and a CVM-MINI network analyzer, which can measure, calculate and display the main electrical parameters, were connected to the machine. This CPS enables the real-time data acquisition of above 100 variables per second, which are stored in the cloud every 15 min. Furthermore, we develop software using Python. This software turns the real-time data into valuable real-time information and indicators, which finally are showed on a dashboard app.

Above the real-time acquired data execution, there are variables such as machine time, number of parts (produced pieces), energy consumption, tool number (number of tool which is being used), tool change (when is the tool changing), and rpm (spindle rotation). The last 4 variables mentioned are used in the execution of this case study to obtain the diary energy consumption per tool in our CPS.

As can be seen in Figure 2 and Table 1, the variable tool number allows us to know how many tools have been used during the day. Once we know the number of tools, the combination of variables energy, tool number, tool change, and rpm spindle give us the energy consumption of each tool per use. Finally, the sum of this consumption for each tool results in the diary energy consumption per tool.

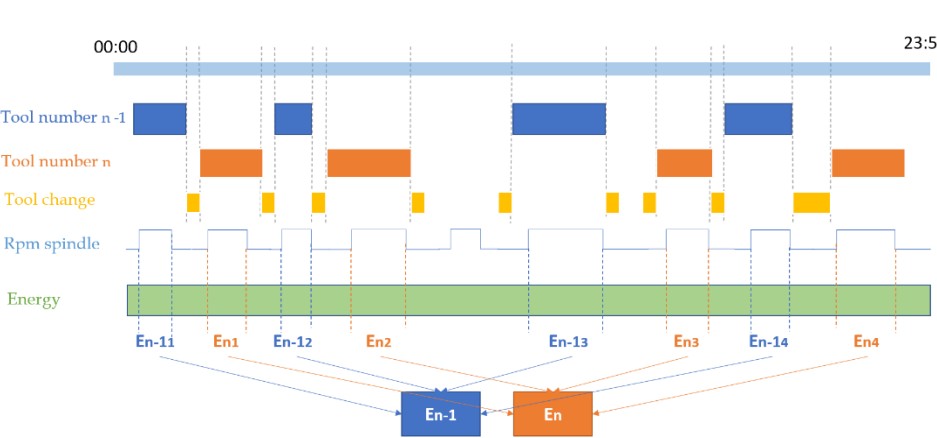

**Figure 2.** Flow chart of the calculations.

**Table 1.** Description of the variables.

| Variable | Description |
|----------|-------------|
| Tool number | It represents the number of each tool which is associated with the tool position in the tool carousel |
| Tool change | It is a counter of how many tool changes have been made |
| Rpm spindle | It shows the rpm of the spindle for each second |
| Energy | It represents the energy consumption (kWh) per second |

One of the main objectives of this research is to highlight the importance of real-time data acquisition and the valuable information. This can be acquired from the implementation of devices and sensors in machines. These kind of calculations are almost impossible to perform in conventional machines, because it is not possible to make a distinction between the actions that consume energy. The CPS allows this type of calculus, not only in terms of energy, but also in terms of time. Therefore, it is simpler with a CPS and a good software development to associate time and energy terms with each machine process in real time. It can be used to calculate existing indicators in real time and with more accuracy, and to develop new indicators that could not be contemplated in conventional machines, because the variables are not available.

In the following section can be seen the results of this implementation.

## 4. Results

This section shows the results of this implementation for one day, in which the machine was machining for 2 h. Figure 3 shows the number tool timeline, which has been used to prove that the developed algorithm detects the tool numbers and the time that each tool is used properly.

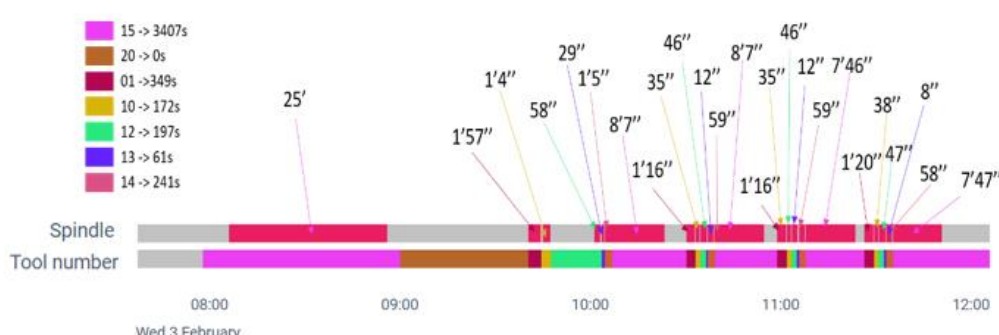

**Figure 3.** Tool number timeline.

These times of use can be used to check our algorithm, which sums up for each tool the time which is really being used, thus the rpm of the spindle is different from zero and therefore, the tool is machining. In Figure 4 can be seen the times obtained from our algorithm, which fit with the sum obtained from Figure 3. The x value of Figure 4 refers to the number of the tool which is being used, because this is how they are numbered, whereas the y value is the time in which each tool has been used.

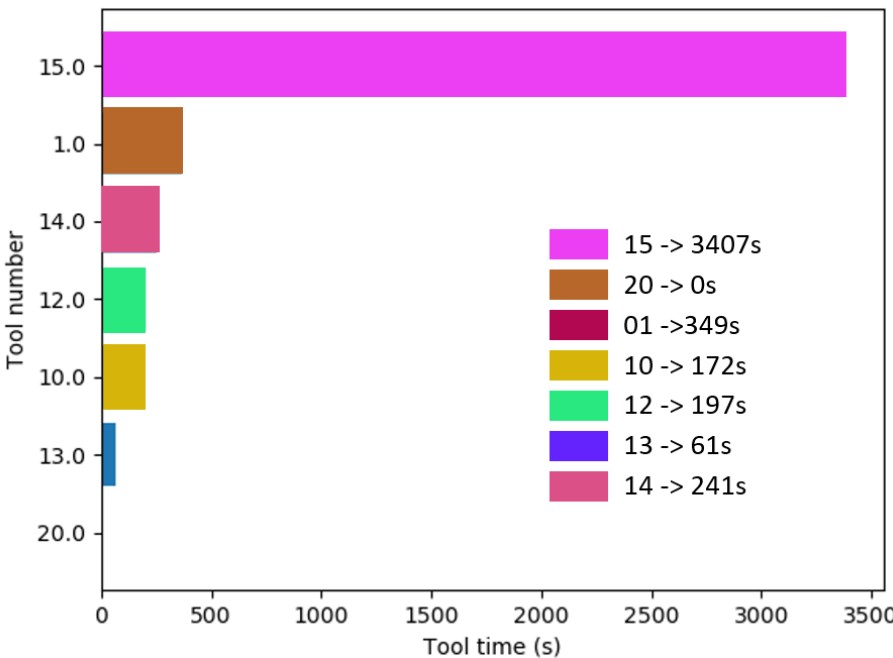

**Figure 4.** Time of use per tool(s).

Once we know when each tool is being used, the energy consumption during these periods of use is collected for each tool, so the developed algorithm prints the total energy consumption, the percentage consumed by tools, and shows the consumption per tool in an orderly manner (see Figure 5). During the day when 2361 Wh has been consumed, 95.55% belongs to energy during the use of tools.

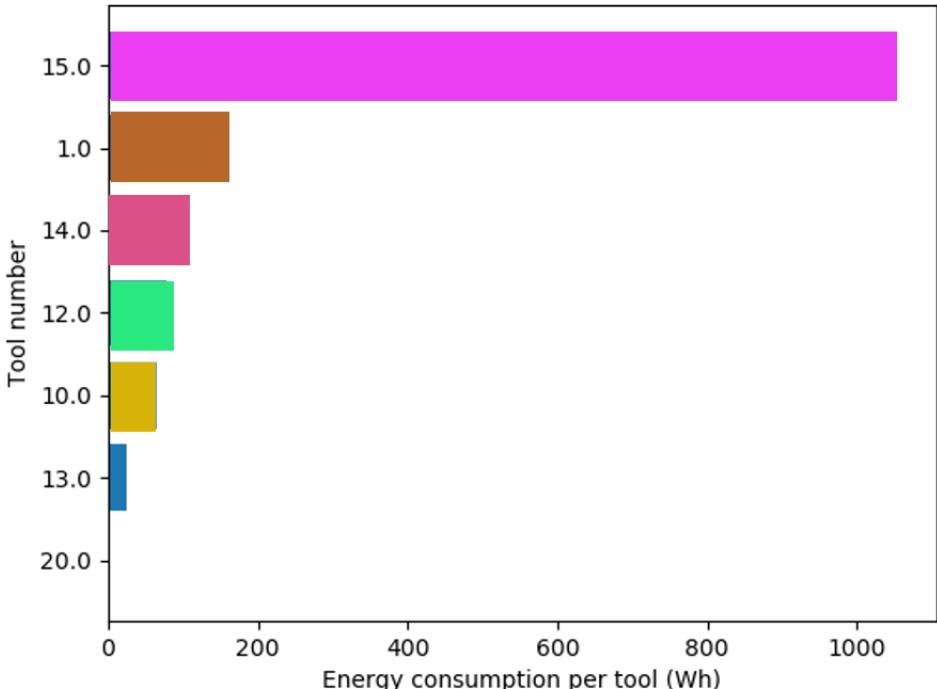

**Figure 5.** Energy consumption per tool.

## 5. Discussion

These real-time calculations enhance the development of predictive maintenance and the analysis of the machine's energy consumption.

The comparison between Figures 4 and 5 shows that time and energy consumption are proportional. Therefore, a disproportionate level of energy consumption in a tool means that this tool is not working right and maintenance for reparison or change of this tool is required. Furthermore, analyzing the percentage of energy consumption—which does not belong to the use of tools, in this case almost 5%—it can be seen if there is a lot of energy consumption losses or, on the contrary, almost all the energy consumption is associated with the machining process, so that the energy efficiency of the machine can be analyzed.

Moreover, knowing the tool life, an alarm can be implemented with the algorithm which calculates the used time per tool (see Figure 4) when the tool life is near to end, helping with the maintenance scheduling. This idea involves a reduction of rejects, since the tool has always changed at the end of its life, avoiding a bad machining process.

## 6. Conclusions

All things considered, it can be concluded that having real-time indicators is crucial to decision-making in the business world [23]. Not only are they important for their immediacy, but also for their accuracy. Thanks to this real-time data acquisition and the algorithm development, maintenance scheduling is easier and variables that are impossible to measure in conventional machines can be included in the development of new KPIs or can accurately determine the calculus of the existed ones. Some developments of new KPIs can be seen in previous researches [11,23].

Therefore, Industry 4.0, particularly CPS, can provide a competitive advantage over the competition, e.g., allowing the use of temporal measurements, which were expensive and complex to measure before it appeared.

To conclude, this research has some limitations, since the results are not yet quantified because it is necessary to define and analyze more case studies. Thinking about a research agenda about this topic and for future research, the development and implementation of real-time indicators should be researched and standardized, since the key for a great decision-making tool is how to transform the acquired data into valuable information through the right KPIs.

**Author Contributions:** Conceptualization, P.M. and M.P.L.; methodology, P.M. and M.P.L.; software, P.M.; validation, M.P.L.; formal analysis, P.M. and M.P.L.; investigation, P.M. and M.P.L.; resources, P.M.; writing—original draft preparation, M.P.L.; writing—review and editing, M.P.L.; visualization, P.M.; supervision, J.A.R.; project administration, J.A.R. and J.C.S. All authors have read and agreed to the published version of the manuscript.

**Funding:** This research received no external funding.

**Conflicts of Interest:** The authors declare no conflict of interest.

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
