# Peer review of "The Importance of Implementing Cyber Physical Systems to Acquire Real-Time Data and Indicators"

_2571-8800, doi:10.3390/j4020012_

Round 1
Reviewer 1 Report
This paper is well written this have objective shows a case study in a Cyber physical system, which is able to acquire the diary energy consumption per tool and to show it to the users. In your structure is necessary improve main those sections Introduction, materials and methods and Discussion. I missed of a section of Related works In the section Materials and Methods is necessary improve detalhed of objective, instrumentation, execution. Is necessary padronize the word FIGURE, because some word aren't with first letter upper case.Author Response
All the changes are explained at the attached PDF.
Thanks for your considerations.

Reviewer 2 Report
Authors present a case study of CPS and discuss the benefit of a real time data acquisition system. Overall the article present some solid work. However, at the same time, I would recommend the following points:
Highlight in the abstract the key novelty/approach/claim of your work
Conclude section 1 with a paragraph that describe how the rest of the paper is organized.
Proposed section 2:
Add a related work section where you mention the work of other researchers that are currently investigating similar problems as well as possible approaches.
In particular I would be curious to know how authors position themselves in the discussion of moving from data (coming from IoT) to information
Here you can find a couple of pointers:
[1] http://dx.doi.org/10.13140/RG.2.2.16339.53286
[2] https://doi.org/10.3390/fi11020036
Current section 2:
- Add a table that describe the variable
Current section 3:
- Color plots in figure 4 following the colors of the one presented). Explain the x and y values of figure 4 and 5
- Describe the algorithm with pseudo code instead of saying “the developed algorithm prints the total energy consumption”
Current section 4:
- Can you quantify the benefit of this approach in term of energy consumption and/or economical benefit for your particular use case?
Author Response
All the changes are explained at the attached PDF.
Thanks for your considerations.

Round 2
Reviewer 2 Report
Thanks for making the changes. I believe that my comments have been addressed in full.
Author Response
Thank you for all your comments to improve our paper